# Update on Distribution and Conservation Status of Amphibians in the Democratic People’s Republic of Korea: Conclusions Based on Field Surveys, Environmental Modelling, Molecular Analyses and Call Properties

**DOI:** 10.3390/ani11072057

**Published:** 2021-07-09

**Authors:** Amaël Borzée, Spartak N. Litvinchuk, Kyongsim Ri, Desiree Andersen, Tu Yong Nam, Gwang Hyok Jon, Ho Song Man, Jong Sik Choe, Sera Kwon, Siti N. Othman, Kevin Messenger, Yoonhyuk Bae, Yucheol Shin, Ajoung Kim, Irina Maslova, Jennifer Luedtke, Louise Hobin, Nial Moores, Bernhard Seliger, Felix Glenk, Yikweon Jang

**Affiliations:** 1Laboratory of Animal Behaviour and Conservation, College of Biology and the Environment, Nanjing Forestry University, 159 Longpan Rd, Nanjing 210037, China; gyyh0303@gmail.com (Y.B.); brongersmai2@gmail.com (Y.S.); 2Amphibian Specialist Group, IUCN Species Survival Commission, Toronto, ON L5A, Canada; jluedtke@amphibians.org (J.L.); lhobin@amphibians.org (L.H.); 3Institute of Cytology, Russian Academy of Sciences, Tikhoretsky pr. 4, 194064 St. Petersburg, Russia; litvinchukspartak@yandex.ru; 4Department of Zoology and Physiology, Dagestan State University, Gadzhiyev str. 43-a, Makhachkala, 3367000 Dagestan, Russia; 5Department of International Economic Cooperation, Ministry of Land and Environment Protection, Pyongyang, Democratic People’s Republic of Korea; 6Interdisciplinary Program of Eco Creative, Ewha Womans University, Seoul 03760, Korea; desireeka93@hotmail.com (D.A.); sera0622@hanmail.net (S.K.); dy.othman@gmail.com (S.N.O.); ajkim129@gmail.com (A.K.); 7Institute of Zoology, State Academy of Science, Daesong-dong, Daesong District, Pyongyang, Democratic People’s Republic of Korea; 8Department of Ecology, State Academy of Science, Daesong-dong, Daesong District, Pyongyang, Democratic People’s Republic of Korea; 9Department of Ecology, Life Science College, Kim Il Sung University, Ryongnam-dong, Daesong-dong, Daesong District, Pyongyang, Democratic People’s Republic of Korea; 10Herpetology and Applied Conservation Lab, College of Biology and the Environment, Nanjing Forestry University, 159 Longpan Rd, Nanjing 210037, China; herpsrule2@aol.com; 11Federal Scientific Center of the East Asia Terrestrial Biodiversity, Far Eastern Branch of Russian Academy of Sciences, 690022 Vladivostok, Russia; irinarana@yandex.ru; 12Re:wild, Austin, TX 78746, USA; 13Birds Korea, 101-1902, Hyundai I Park, Busan 48559, Korea; nial.moores@birdskorea.org; 14Hanns Seidel Foundation, Seoul 04419, Korea; bjseliger@yahoo.de (B.S.); glenk@hss.de (F.G.); 15Department of Life Sciences and Division of EcoScience, Ewha Womans University, Seoul 03760, Korea; jangy@ewha.ac.kr

**Keywords:** anuran, caudata, extinction risk assessment, landscape modelling, molecular identification, Northeast Asia, salamander, toad, treefrog

## Abstract

**Simple Summary:**

In this study, we used field and literature surveys, call recordings, photographs, landscape models and molecular tools to estimate the presence, range and status of amphibians in the Democratic People’s Republic of Korea. We found 18 native species and the suspected presence of *Karsenia koreana* and two *Onychodactylus* species. We also determined northern range boundaries for *Rana uenoi* and *Dryophytes japonicus* with molecular tools. Based on distribution and modelling, we can expect the contact zone between species within the *Rana* and *Onychodactylus* genera to be located along the Changbai Massif, a high mountain range. The species richness was higher in the lowlands and at lower latitudes, with up to 11 species, while species richness in northern areas was half that value. Following the categories and criteria of The IUCN Red List of species, ecological models and known threats, we recommended ten species to be listed as threatened at the national level. The ecology of species in the DPR Korea is understudied, although species reliant on agricultural wetlands (e.g., rice paddies) are not as threatened as species living in forested areas due to the enduring presence of extensive agricultural landscapes.

**Abstract:**

Determining the range, status, ecology and behaviour of species from areas where surveys and samplings are uncommon or difficult to conduct is a challenge, such as in the Democratic People’s Republic of Korea (DPR Korea). Here, we used genetic samples, field surveys, call recordings, photographic identification and a literature review to estimate the presence, range and status of amphibians in the DPR Korea. From our combined results and based on the IUCN Red List categories and criteria, we were able to estimate the national threat levels for most species. Our results demonstrated the presence of 18 native species and the suspected presence of *Karsenia koreana* and two *Onychodactylus* species. We reported the first record for *Rana uenoi* in the vicinity of Pyongyang using molecular tools and similarly confirmed the presence of *Dryophytes japonicus* at the same location. Based on distribution and modelling, we can expect the contact zone between species within the *Rana* and *Onychodactylus* genera to be located along the Changbai Massif, a mountain range that marks a shift in ecoregions and acts as a barrier to dispersion. The species richness was higher in the lowlands and at lower latitudes, with such areas populated by up to 11 species, while more northern regions were characterised by species richness of about half of that value. The combination of ecological models and known threats resulted in the recommendation of ten species as threatened at the national level following the IUCN Red List categories and criteria. This high number of threatened species was anticipated based on the high threat level to amphibians in bordering nations and globally. While the ecology of species in the DPR Korea is still understudied, we argue that species relying on agricultural wetlands such as rice paddies are not under imminent threat due to the enduring presence of extensive agricultural landscapes with low rates of chemical use and mechanisation. The maintenance of such landscapes is a clear benefit to amphibian species, in contrast to more industrialised agricultural landscapes in neighbouring nations. In comparison, the status of species dependent on forested habitats is unclear and threat levels are likely to be higher because of deforestation, as in neighbouring nations.

## 1. Introduction

Delineating the boundaries of species distributions is a preliminary requisite to determining potential presence, taxonomic research and threat levels that may potentially require conservation actions. The distributions of large vertebrates such as mammals and birds are now coarsely defined for most species, but details are missing for a large number of species from other taxonomic groups [1]. For instance, as of March 2019, 21.35% of described amphibians (1443 species) were still classified as data-deficient on the IUCN Red List of threatened species, a reasonable proxy for missing distribution data when in conjunction with a handful of other variables. Many of these amphibian species are from areas where it is difficult to collect data. This is the case for the Democratic People’s Republic of Korea (hereafter, DPR Korea), where recent scientific exploration and research has not been as intense as in many other nations. Some exceptions do exist, however, especially for birds [2,3,4]. Presence points for some amphibian species in DPR Korea are available from non-focal references, such as wetland inventories, but these do not constitute descriptions of distribution ranges by themselves [5,6,7,8].

Fifteen species are listed as present in DPR Korea in the update by Kim and Han [9], and a sixteenth invasive species, *Lithobates catesbeianus*, is also listed [10]. While most of the species are valid [11], phylogenetic developments have shown that *Rana ornativentris* [12,13], *R. temporaria* [14] and *R. chensinensis* [15,16,17] do not occur in DPR Korea. In addition, *Bufo sambangensis* has not been studied since its description in 1996, and the status of the species as well as its potential synonymy with *B. stejnegeri* [18] have not been confirmed through molecular tools [19]. Similarly, *Dryophytes suweonensis* (following [20]; previously *Hyla suweonensis* [21]) is missing from the list [22,23], while other genera such as *Onychodactylus* and *Karsenia* are expected to be present, although their presence in the nation has not yet been confirmed [24,25,26].

In addition, the absence of data on ranges is especially true for amphibians, as they are generally harder to detect and are sometimes considered to be of lower interest than other groups of species [27,28]. Combined with DPR Korea’s geography, i.e., being flanked by sea both to the east and the west, this means that the distribution of species in the nation is especially difficult to estimate. Species present both to the south and to the north of DPR Korea are expected to be present throughout the nation; however, when a species is present only to the north or the south, the boundaries of such ranges are unknown. For example, the range of *Pelophylax chosenicus* [29] was expected to extend into DPR Korea, as it is known to occur very close to its southern border [30,31,32]. Accordingly, a recent survey found the species substantially further north than originally expected [11,33].

In addition, the fact that a species occurs both to the north and to the south of DPR Korea does not necessarily make its distribution easier to predict. For instance, *D. japonicus* [34] is expected to be present in most of the nation [9] as it occurs both to the north and to the south [35] and because the species’ ecological requirements can be met throughout the majority of the area [36]; however, *D. japonicus* is divided into several clades [37] and the southern distribution limit of individuals assigned to the clade, referred to as the “*ussuriensis* clade” by some authors [38,39], remains unknown.

A geographically similar situation involving several species arises for the *Rana* genus where the range of multiple species borders the DPR Korea. While some species are present only to the south (e.g., *R. uenoi* [40]) or to the north (e.g., *R. dybowskii* [41,42], *R. amurensis* [43,44] and *R. chensinensis* [45]), others such as *R. huanrenensis* [46,47,48] and *R. coreana* [45,49] are present both to the north and south. Consequently, some of the species present in the Primorsky Krai in the Russian Federation (hereafter Russia), in the north-eastern People’s Republic of China (hereafter PR China) and along the northern border of the Republic of Korea (hereafter R Korea), may be present in DPR Korea. In this case, the species are likely in contact and potentially hybridising. In theory, and with the understanding that habitat segregation between *Rana* species is a potential factor for speciation, clarifying the presence of species through molecular tools should be straightforward if genetic samples are available (e.g., [9,50,51,52]).

Here, we combine the use of molecular tools on old museum collections and our field surveys to determine the distributions of all species occurring in the nation and to assess their conservation status where possible. This project relies on several types of data and analyses—first mitochondrial genetic data from historical specimens, followed by photographic identification of field and museum animals, aural and visual field surveys and acoustic analyses. Finally, we assess species diversity, model species ranges and provided a preliminary national conservation status for all amphibian species in DPR Korea. The last steps are conducted in order to prioritise conservation targets, both for species and habitat types, which would benefit a large range of species. We expect several species to be under higher levels of threat, in line with the current amphibian crisis [27,53,54], while other species are expected to be widespread and found in large numbers.

## 2. Materials and Methods

### 2.1. Taxonomic Sampling

The genetic samples used in this project were collected at an unspecified location in the vicinity of Pyongyang, DPR Korea (Figure 1). The samples used here are muscle tissues from *Rana* sp. (*n* = 13) and *Dryophytes* sp. (*n* = 12). The *Rana* sp. collection was conducted by the North Korea Academy of Sciences (Chosun Academy of Science) and samples (voucher IDs in Table 1) were sent to the ZISP (Zoological Institute of the Russian Academy of Sciences, St. Petersburg, Russia) during “Soviet times”, prior to the 1990’s. The *Dryophytes* sp. collection comes from two sources. Eight adult specimens have the same origin as the *Rana* sp. samples. An adult and two juvenile specimens were collected in July 1947 and housed in the collection of the S.M. Kirov Military Medical Academy (St. Petersburg, Russia) until 2018, after which they were transferred by I.V. Doronin to the ZISP collection. Three of the *Dryophytes* samples were stored in formalin, while all other *Rana* and *Dryophytes* samples were stored in alcohol. Although it is unknown how the samples were fixed, we suspect fixation to have been conducted in formalin based on the sampling period. Consequently, fixation and dehydration were unlikely to have been conducted following current recommendations, such as fixation times of 14–24 h and thorough dehydration prior to embedding in order to avoid severe DNA fragmentation.

### 2.2. Genetic Identification

Genetic identification of individuals was needed when target species did not present discrete morphologies, such as between *R. uenoi* and *R. dybowskii*. Misidentifications are especially problematic when estimating distributions and generate inaccurate results [56], potentially overestimating or underestimating the threat levels to species.

Advancements in technologies have made it possible to extract ancient DNA (e.g., [57]) or DNA from biological remnants such as insect exuviae [58]. Similarly, it is possible to retrieve DNA from samples fixed and stored in formalin [59] or other fixation chemicals [60]; however, this task is made increasingly difficult when samples are fixed in an unknown chemical and stored for decades in another chemical with unknown intermediate treatment.

For this reason, we adapted the Qiagen Formalin DNA extraction protocol (QIAamp^®^ DNA FFPE Tissue; Venlo, The Netherlands). Here, not all samples were entirely lysed within 3 h, already 2 h longer than described in step 11 of the manufacturer standard protocol [61]. We avoided longer incubation times or higher incubation temperatures as these may have resulted in more fragmented DNA; thus, we removed the pellet 3 h after the beginning of incubation as described in step 11 and restarted the procedure from step 3 with the pellet extracted, i.e., for the remaining tissue. This multiplication of steps was conducted so that tissues already lysed were not submitted to further degradation from heat. All resulting samples were treated independently.

Once the genomic DNA was extracted and eluted, we used it for barcoding with the primers and protocols described in the literature [42,62,63]. All PCRs were run in duplicate and as gradient PCR, with a two-degree increment framing the recommended annealing temperature in two steps, such as −4 °C, −2 °C, Ta, +2 °C and +4 °C. For all samples, we amplified a fragment of the *12S* ribosomal small subunit. For *Dryophytes* sp. samples, the primer pair and protocol used followed [62], while for *Rana* sp., the primer pair and protocols were the ones used in [42]. PCR reactions were performed in a SimpliAmp™ Thermal Cycler (Applied Biosystems, Waltham, MA, USA). Products were visualised on 1.5% agarose gel loaded with three microliters of PCR products, run on an Agaro-Power™ System (A-7020, Bioneer, Daejeon, R Korea) and visualised with a Nucleic Acid Bioimaging Instrument Blue Illuminator (S, NeoScience, Suwon, R Korea) using TopGreen Nucleic Acid 6× Loading Dye (GenomicBase, Seoul, R Korea).

Most PCRs did not yield visible bands, but all samples with a single band visible were sent for sequencing. When multiple bands were visible, the band with the target product size only was extracted (FavorPrep Gel/PCR purification mini kit; Favorgen, Taiwan) and sent for sequencing. All samples were sent for purification and both forward and reverse sequencing by Cosmogenetech (Cosmogenetech Co., Ltd., Seoul, R Korea).

The resulting sequences were trimmed and analysed in Geneious v 11.0.2 (Biomatters Limited, Auckland, New Zealand). Only two sequences for *Dryophytes* and eight sequences for *Rana* were good enough to be retained. To assess which clades the samples belonged to, we imported georeferenced sequences from GenBank (Table 2), based on the closest match when blasted against the NCBI database and matching sequences for all species listed [9]. The missing georeferences for GenBank samples were extracted from the literature [64,65,66,67,68] for *Dryophytes* sp. and from [42,48,69,70] for *Rana* sp.

The sequences were aligned independently for each genera using MUSCLE (v3.8.31; [71]), implemented through the Geneious plug-in, with a maximum of 10 iterations following default parameters and further revised manually when needed. We then reconstructed a maximum likelihood (ML) tree for reach genera, inferred from the *12S* fragment and 349 bp long for *Dryophytes* sp. and 293 bp long for *Rana* sp. We also added *Hyla orientalis* as the outgroup for the *Dryophytes* sp. dataset and *R. kukunoris* as the outgroup clade for the *Rana* sp. dataset (GenBank accession numbers shown in Table 2). All analyses were performed using the PHYML plugin in Geneious [72]. This method implements a fast and accurate heuristic estimate of maximum likelihood phylogenies, with GTR selected as the substitution model for flexibility reasons. We performed the run with 50,000 bootstraps under default variables.

### 2.3. Pictures Identification

Among the individuals that could not be identified by molecular markers, some could be identified through morphological cues. *Rana coreana* [73] and *R. amurensis* have warts on their lateral side, which are usually black [44] but sometimes red in *R. amurensis* during the breeding season [74,75]. In contrast, these warts are absent on *Rana* samples not pertaining to the *R. amurensis* group [76], specifically *R. chensinensis, R. pirica* [69], *R. huanrenensis, R. dybowskii* and incidentally *R. uenoi* here (see [40,45]).

For hylids, *D. suweonensis* is smaller and more slender than *D. japonicus*, and the angle between the eyes and ipsilateral nostrils can be used for species identification [77]. Although morphologically differentiating traits with *D. immaculatus* and *D. flaviventris* are present [78], the species is not present in the area [79] and *D. suweonensis* is known to be present further north than Pyongyang [22,23].

### 2.4. Field Surveys

Field surveys to detect species presence were conducted in 2016, 2018 and 2019. The surveys conducted in Mundeok (39.5489° N, 125.4390° E; datum = WGS84) in March 2017 and May 2018 were mostly opportunistic, as they were originally focused on avifauna, but the area was surveyed again in June 2019 with a specific focus on amphibians. Opportunistic surveys were also conducted in Kumya, South Hamgyong (39.5386° N, 127.2206° E) in late May 2018. Finally, amphibian-focused surveys were conducted between 24 and 26 March and between 2 and 6 June 2018 in the area of Rason, within the localities of Rajin and Sonbong, and limited north by the Tumen River, marking the border between DPR Korea and Russia (42.3446° N, 130.4712° E; Figure 1). This coastal area is characterised by high hills in the south and wetlands in the north, including a Ramsar site, the Rason Migratory Bird Reserve. The forested habitat is substantially, degraded except for isolated patches; however, agricultural wetlands are numerous and most of the agricultural work is conducted by hand, providing adequate substitute wetlands for amphibians.

During amphibian-focused surveys, each time a habitat potentially hosting amphibians was encountered along a predetermined route, the site was investigated through aural and visual surveys for amphibian presence. Surveys were conducted through aural and visual detection, following established protocols whenever possible [36,80,81]; however, surveys were required to avoid encounters with local people and protocols were adapted following this and other local requirements. Daytime surveys were conducted under general search-and-encounter protocols, including substrate flipping when allowed. Night-time surveys followed spotlight and call count protocols [36,82]. We also took ecological measurements, whereby we measured the water quality with a PCSTestr 35 multimetre (Oakton Instruments; Vernon Hills, IL, USA) for salinity (ppm), pH, temperature (°C), conductivity (μS) and total dissolved solids (tds; ppm). Several types of landscapes were investigated, including natural wetlands, agricultural wetlands, grassy hills and forested areas.

In addition, snout–vent–length (SVL) measurements were performed down to the nearest 0.1 mm for ten individual *D. japonicus* samples caught in a rice paddy close to Rajin (42.304538° N, 130.390762° E; callipers 1108-150 W, Insize; Suzhou, China). The SVL values for these 10 individuals were compared through a *t*-test (SPSS, v21, IBM SPSS Statistics Inc., Chicago, IL, USA) to the SVL values of 308 individuals measured in R Korea, PR China, Mongolia, Russia and Japan (data for individual frogs outside of DPR Korea not provided here) to characterise the variation within the species.

### 2.5. Call Properties

Five acoustic files between 31 and 63 s in length were obtained from Mundeok (39.5489° N, 125.4390° E) in May 2017. The acoustic files were recorded in MP3 format, which was converted into WAV format using VLC media player v. 3.0.4. The files all had high levels of background noise, which was reduced using Audacity^®^ (v2.2.0). Data were then extracted using R [83] and the acoustic packages WarbleR [84], seewave [85] and tuneR [86]. We then described call properties based on oscillograms and spectrograms and measured call properties such as fundamental and dominant frequencies, before comparing them to calls from other parts of the species range [87].

### 2.6. Distribution Estimates and Species Richness

To model distribution estimates for all species, we used the datapoints presented above and investigated all accessible literature to assess the presence of all amphibian species expected to be extant in DPR Korea. To date, the most extensive and accurate studies on the subject were the ones conducted by [88,89], which were mostly transcribed by [51] and completed by [9]. The presence points are reported on Figure 1 and GPS coordinates are reported in Appendix A. All studies citing amphibians from DPR Korea available were analysed, although some references only reported the potential presence of the species or did not provide clear georeferenced datapoints associated with specific species [6,18,29,90,91,92,93,94,95,96]; thus, only clear and georeferenced presence points are listed below. Several presence points were acknowledged by [11] and used as test data for the accuracy of the data collected. In addition, datapoints available from the North Korean Human Geography website (http://www.cybernk.net (accessed on 1 July 2019)) were directly extracted from, meaning the reference was not added to the list below. *Bufo sambangi* [19], also spelled *B. sambangensis* [9,10], was not included in this analysis due to the paucity of information supporting the existence of the species and morphological similarities with *B. stejnegeri* (presented by [9]; Figure 2). Additional datapoints were provided from past field observations during a workshop in Pyongyang in June 2019, but only 11 were retained as they were within 20 km of a confirmed presence point.

Furthermore, all georeferenced mentions of *R. uenoi, R. dybowskii*, *R. temporaria, R. chensinensis, R. ornativentris* or any of the combination above using either species name as species or subspecies were assigned to the corresponding clade based on the recent nomenclature when available, or were not used in models when no consensus could be reached (although the GPS data are presented in Appendix A under the label *Rana* sp.). Morphological cues to discriminate between *R. uenoi* and *R. dybowskii* have not yet been described and the range of the two species is expected to be continuous, while the location of the contact zone between the two species has still not been determined. *Rana huanrenensis* is not included in any studies from DPR Korea, but data from the R Korea and PR China were used to model the range of the species. Finally, we considered *Onychodactylus* to be present in DPR Korea based on [24], and considered *O. zhaoermii* as the species with the largest range in northern regions and *O. zhangyapingi* as potentially present; however, neither of the species has been recorded under those names in the nation yet. The datapoints used in the models are listed below.

#### 2.6.1. Anura

*Bufo gargarizans* [97]: [9,50,76,89,91,98,99]; *Bufo stejnegeri*: [5,100,101]; *Strauchbufo raddei* [102]*:* [7,51,89]; *Dryophytes japonicus*: [5,9,50,51,91,98,103,104]; *Dryophytes suweonensis*: [22,23]; *Bombina orientalis* [105]: [9,50,51,52,73,91,98,100,106,107,108,109,110]; *Kaloula borealis* [111]: [9,50]; *Rana amurensis:* ZISP collection, [73]; *Rana coreana* and *R. amurensis*: [9,50,73,107,112,113,114,115]; *Rana uenoi* and *R. dybowskii*: [50,51,73,91,107] (using identification cues from [116] and integrating the work from [88,89]) [5,9,117]; *Glandirana emeljanovi* [118]: [9,50,51,73,91]; *Pelophylax nigromaculatus* [119]: [50,51,73,98,99,100,107,120]); *Pelophylax chosenicus*: [9,33].

#### 2.6.2. Caudata

*Hynobius leechii*: [9,50,99,100,121,122]; *Onychodactylus koreanus*: [50,51] (originating from [89]) [5,7]; *Salamandrella tridactyla*: [51] (originating from [89]) [9,112,123].

It should be noted that three references of potential interest, as cited in other work, could not be traced, namely [124,125]. These were not included in our models, despite the potential presence of valuable datapoints. For our models, we also used the datapoints from GBIF.org (doi.org/10.15468/dl.qetyb9 (accessed on 26 February 2019)), filtered for amphibians from DPR Korea, R Korea, PR China and Russia. The non-focal species of this selection were filtered out, and in total we used 24,193 data points for all species confounded (Appendix A for all datapoints for all species; Appendix A for datapoints within DPR Korea). Historical datapoints from [126] were not used for the models. Additional fine tuning for specific species was also conducted.

*Rana coreana*: The northernmost point in Ryanggang [107] was excluded as the morphological cues to discriminate between *R. coreana* and *R. amurensis* had not yet been determined at that time.

*Rana uenoi* and *R. dybowskii*: The cut-off point used for the range of these two species was the latitude matching with the locality closest to Pyongyang, based on the genetic analysis conducted in this study.

*Rana huanrenensis*: We added georeferenced datapoints from the literature [48,127,128] to avoid north–south bias in the number of points.

*Onychodactylus* sp.: Due to the unknown range boundaries of all four species potentially present in DPR Korea and the absence of data, a single model was created, including all records for the genus (including additional datapoints from [24,25]). Only *O. fischeri* has been listed from DPR Korea so far, despite its presence being unlikely or limited to the extreme north-east of the nation. As *O. koreanus* is the only species for which sufficient confirmed datapoints were available for an independent species model, we ran two models, one for *Onychodactylus* sp. and one for *O. koreanus*.

*Salamandrella tridactyla*: Additional datapoints from [25,129] were used for the models.

*Karsenia koreana*: This species has not been found in the nation yet and was only recently described [130]; it may be present in the south-east of DPR Korea [26]. Due to the paucity of information, we did not include this species in the list of modelled species.

*Lithobates catesbeianus*: The presence of the species has been confirmed [10] in DPR Korea, but we did not include models for the species as it is invasive.

We used maximum entropy modelling (MaxEnt; [131]) of 19 bioclimatic factors (Hijmans et al., 2005: percent contribution, permutation importance and training gain for each bioclimatic variables in Appendix A Appendix A, respectively) to delineate potential ranges of each amphibian species. We decided to include all variables despite the potential correlations, as ecological requirements are not known for the species selected here and they are bound to differ greatly. A subselection of variables would result in less accurate models for some of the species, as well as an inability to compare variable responses among species; therefore, to avoid the exclusion of relevant variables [132], but also to create a baseline for future studies, we decided to include all bioclimatic variables and to ensure the absence of preconceived bias on the relevant ecological variables. For each species and model, MaxEnt was trained for ten bootstrap replicates with 20% random test percentage and then projected to 30 arc second (0.00833 dd) rasters of the Korean Peninsula. To balance the sampling bias generated by more numerous datapoints in R Korea compared to DPR Korea, we constructed two bias layers, namely 1 (DPR Korea):10 (R Korea) and 1 (DPR Korea):5 (R Korea)—equivalent to the distribution of collected unique occurrence points. Additionally, duplicate presence records (presences within the same grid cells of the environmental layers) were removed. For each species, the model selected was based on the highest true skill statistic (TSS) of the 10-percentile training omission threshold (Table 3). We modified models further for *R. amurensis* and *R. huanrenensis*; the models were trained with 0.02 dd rasters of Northeast Asia (without bias layer) and projected to 0.00833 dd rasters of the Korean Peninsula to remedy the insufficient presence of data points in DPR Korea. Predicted presence was determined with the following thresholds: minimum presence, 95% presence, 10-percentile training omission. The 10-percentile training omission threshold was further used for TSS reporting [133]. The total surface areas of the minimum presence and 10-percentile training omission polygons were calculated as each species’ expected land surface area. We also compiled a species richness map for the all species merged together for DPR Korea, using a sum of all presence probabilities for species distribution models, where richness values were classified as 0 (not suitable/present), 0.25 (minimum presence), 0.5 (95%) and 1.0 (10-percentile training omission).

### 2.7. Extinction Risk Assessment

One of the most common and most reliable ways to conduct assessments of extinction risk is through the use of the IUCN Red List categories and criteria [134]. These have been used to inform and catalyse the conservation of numerous species [135,136]. The data on the IUCN Red List are used in conjunction with other tools to determine conservation priorities (e.g., [137,138]), the identification and delineation of protected areas (e.g., [139]) and as an indicator of the health of the planet’s biodiversity [140]. The IUCN Red List categories and criteria are applied at the global scale [141], but they have also been adapted for national [142] and regional scales [143,144].

IUCN requires that a species be evaluated against the quantitative thresholds of the five criteria that determine the risk of extinction. When a species meets one or more of the criteria, it is assigned to one of the three threatened categories based on the threshold that is met: vulnerable, endangered or critically endangered. The criteria are: A, population size reduction; B, geographic range size; C, small population size and decline; D, very small population or restricted distribution; E, quantitative analysis of extinction risk. A species that does not meet the threshold for any of these criteria is placed into one of the other IUCN categories; however, as very few animal populations are monitored regularly and in their totality [134], protocols have been established for small datasets, which can encompass a broad type of data and account for varying degrees of uncertainty in the event that no robust data are available [145,146].

Although some authors have provided estimates of population sizes (Table 4), and others have provided data on the type of threat to be considered for specific species, population dynamics in DPR Korea are mostly unknown. We conducted a workshop in Pyongyang in June 2019, during which 12 herpetologists with knowledge on species within DPR Korea were asked to provide population trends for the species they knew about (Table 4). From our own surveys, no exact numbers can be reported due to restrictions in field surveys, but the assessment is a comparison with known population sizes in PR China, Russia and R Korea.

*Dryophytes japonicus*: Population surveys in Rason, Kumya and Mundeok (Figure 1) recorded large population sizes at all sites.

*Dryophytes suweonensis*: Population surveys in Mundeok (Figure 1) recorded a large population size.

*Pelophylax nigromaculatus*: Population surveys in Rason, Kumya and Mundeok (Figure 1) recorded large population sizes at all sites.

*Pelophylax chosenicus*: Population surveys in Mundeok (Figure 1) recorded a large population size.

When population size estimates were not available, extinction risk evaluations were conducted according to estimates of geographic range size (criterion B). The IUCN Red List categories and criteria include two different measures applied to the geographic range. One is the extent of occurrence (EOO), defined as the area contained within the shortest continuous imaginary boundary that can encompass all the known, inferred or projected sites of present occurrence of a taxon, excluding cases of vagrancy, and is a measure of the spread of extinction risk across a species’ range. This requires the delineation of a species’ range, which we modelled in the section above. The second is the area of occurrence (AOO), defined as the area within the EOO that is occupied by the species, which cannot be used here as it requires fairly comprehensive field surveys to generate sufficient occurrence data, which are then assigned to a 2 × 2 km grid cell. Process-based modelling using small-scale environmental variables (e.g., microclimate) can be applied to refine the species distribution models and better determine the possible geographic ranges of species. As an example, here the range of *D. suweonensis* can be modelled based on datapoints available both within and outside of DPR Korea (see species section) and the distribution model can be further restricted based on the maximal elevation known for the species [36]; therefore, we used the result of the MaxEnt models to assess the extent of occurrence (EOO) and assess threat levels following IUCN Red List thresholds and classification schemes, following the guidelines and examples from [134] and threat definitions [147]. The preliminary extinction estimates presented here are guidelines and include all species present and likely to be present in DPR Korea, set against the Red List Regional Guidelines.

## 3. Results

### 3.1. Genetic Identification

Out of the 12 *Dryophytes* sp. samples from which DNA was extracted, lysis was incomplete within the original 3 h for all but four samples. We transferred the four samples to a new lysis buffer and they were totally lysed on the second round. This additional step resulted in a total of 16 samples for which DNA was extracted. The DNA concentration was <0.2 ng/µL for all resulting samples, but we used all of the samples as templates for subsequent PCRs. PCRs and sequencing of adequate quality for the analysis conducted here were successful for only two samples (Appendix A) out of the 160 PCRs conducted (16 samples ran in duplicate on gradient PCR with five temperature increments). The DNA concentration obtained was slightly higher for *Rana* sp. in comparison with *Dryophytes* sp. Out of 13 samples, seven were duplicated for a second round of lysis, resulting in 20 samples with DNA concentrations <0.5 ng/µL, leading to 200 PCR reactions. Counting samples from both genera, seven samples from the initial lysis were successfully sequenced, while three were successfully sequenced from the secondary lysis (Appendix A). Successful sequencing arose from a single PCR at the recommended annealing temperature, while five samples resulted in successful sequencing for recommended Ta + 2 °C and four samples resulted in successful sequencing for recommended Ta + 4 °C (Appendix A). All sequences presented here have unresolved calls in sequencing that do not prevent species identification but would not be adequate for analyses further than the one presented. Consequently, sequences were not submitted to GenBank due to further need for protein annotation but are presented in Appendix A.

The reconstructed ML tree showed that the two *Dryophytes* samples clustered with *D. japonicus* (ZISP.14002 and 14004; Figure 3). The analysis for *Rana* sp. showed that all samples collected were clustered with *R. uenoi* (ZISP.13968–13981; Figure 4) and are the first report of the species in the vicinity of Pyongyang and in DPR Korea. For this analysis (see Figure 4), the species identification for *Rana dybowskii* individuals from R Korea was arbitrarily reassigned to *R. uenoi* based on [40]. The analysis presented here is adequate in that it depicts *R. dybowskii* and *R. pirica* as sister clades, and highlights the fact that none of the sample for which DNA was extracted belong to the species named in [9].

### 3.2. Picture Identification

From the collection at the ZISP (number 14036) and based on available morphological cues (Figure 5), we identified one individual as belonging to either *R. amurensis* or *R. coreana*, the first record in the vicinity of Pyongyang for either species. In addition to the presence of warts on the lateral sides, the examined individual had a non-interrupted white upper lip (Figure 5A), unlike other Northeast Asian *Rana* sp. We also identified individuals as *D. suweonensis* (ZISP.14019–14021; Figure 5B), *Kaloula borealis* (ZISP.14038–14044) and *Pelophylax nigromaculatus* (ZISP.14022–14033), the first records for these species in the vicinity of Pyongyang.

### 3.3. Field Surveys

Aural field surveys in Mundeok in 2017 resulted in the detection of *D. japonicus*, *D. suweonensis*, *P. nigromaculatus*, *P. chosenicus* and *Bombina orientalis*. The same species, with the exception of *B. orientalis*, were detected in 2018 and 2019. Surveys in Kumya in 2018 only detected *P. nigromaculatus* and *D. japonicus*.

The surveys conducted in Rason in March 2018 resulted in the detection of a single *Rana* sp. egg clutch, which could not be closely investigated and was, thus, assigned to *Rana* sp. The absence of detection of other species does not indicate their absence and may be related to other variables such as weather. During the surveys, the average air temperature was 13.42 ± 2.42 °C (mean ± SD), the average air relative humidity was 43.26 ± 6.01% and the average water temperature was 8.48 ± 4.07 °C (min = 2.10 °C). Based on surveys conducted in R Korea, P.R. China and Russia, the air and water temperatures and air humidity were adequate for spawning by early-breeding species such as *Rana* sp.*, Hynobius* sp*., Salamandrella* sp. and *Onychodactylus* sp. The temperatures were slightly lower than that at which *B. gargarizans* and *Strauchbufo raddei* usually breed and too low for other amphibian species potentially present in the area to be active. The other environmental variables collected, namely salinity, pH, conductivity and total dissolved solids, were also within the values accepted by the species suspected to be present [148,149,150]. The only variable with slightly lower values than otherwise expected was total dissolved solids (120.87 ± 74.01 ppm), likely due to recently melted snow.

The surveys in Rason in June 2018 resulted in the detection of numerous species, including all anuran species expected to be present in the area based on models described below, with the exception of caudata, in a wide variety of wetlands (Figure 6). The water quality was not generally different to that used by the same species at similar latitudes elsewhere. At the coastal site where *S. raddei*, *P. nigromaculatus* and *B. orientalis* were found, the variables were (14:30 hh:mm) air temperature = 30.6 °C, relative humidity = 32.9%, air pressure = 1007.8 hPa, water temperature = 29.2 °C, conductivity = 125.4 µS, pH = 8.04, salinity = 171 ppm and tds = 343 ppm. At one of the agricultural wetlands where *D. japonicus* and *P. nigromaculatus* were recorded, the variables were (20:20) air temperature = 17.5 °C, relative humidity = 71.9%, air pressure = 1008.3 hPa, water temperature = 22.1 °C, conductivity = 203 µS, pH = 8.42, salinity = 145 ppm, and tds = 102 ppm. On the grassy hills where *B. orientalis* was found to breed in small water holes, the variables were (17:00) air temperature = 18.4 °C, relative humidity = 69.3%, air pressure = 1009.4 hPa, water temperature = 31.4 °C, conductivity = 146 µS, pH = 8.91, salinity = 43.4 ppm and tds = 69.3 ppm. In the natural wetland where *B. orientalis* was found (11:00), the variables were air temperature = 27.4 °C, relative humidity = 71.0%, air pressure = 1007.8 hPa, water temperature = 29.2 °C, conductivity = 85.54 µS, pH = 8.89, salinity = 49.3 ppm and tds = 62.7 ppm. Finally, at the natural wetlands where *D. japonicus* tadpoles were found, the environmental variables were (15:40) air temperature = 24.2 °C; relative humidity = 71%, air pressure = 1007.8 hPa, water temperature = 29.2 °C, conductivity = 85.5 µS, pH = 8.89, salinity = 49.3 ppm and tds = 62.7 ppm. GPS coordinates for all sites are provided in Appendix A and are visualised in Figure 6.

*Dryophytes japonicus* was the most commonly detected species at night via mating calls. Despite some instances of individuals calling before sunset and as early as midday, the large majority of individuals called after sunset and choruses were characterised by a calling index equal to three for all populations surveyed after sunset [36]. Tadpoles were also detected in a variety of water bodies, although not at a Gosner stage >25 [104,151], likely due to the early season. The species was widespread in all types of wetlands and present in large numbers in agricultural wetlands. When comparing SVL values of individuals caught in Rason (*n* = 10) to those from neighbouring countries (*n* = 308), the differences were not significant (*t*-test; *p* = 0.063). The SVL value for *D. japonicus* in Rason was 34.62 ± 4.73 cm (mean ± SD).

*Pelophylax nigromaculatus* was the second most abundant species based on call surveys and the most abundant species based on daytime encounter surveys. Tadpoles were, however, not present in large numbers, indicating a delayed breeding season compared to *D. japonicus*. This species was also widespread in all types of wetlands and was present in large numbers in agricultural wetlands.

*Bombina orientalis* was detected at three different sites in densities comparable to those of populations found in R Korea and Russia (*n* < 10). The species was detected through it spawn at one site, best described as grassy hills with tire tracks on a non-asphalted road, which were filled with water. The species was present in small and shallow water bodies but was not found in agricultural wetlands.

*Strauchbufo raddei* tadpoles were detected in a shallow wetland in a sandy landscape close to the sea. *Bufo gargarizans* was detected once only, in a mountainous area during a short survey, and consequently expected to be locally abundant in this habitat. *Rana dybowskii* was also detected once in the same area as *B. gargarizans*, and the species is consequently equally expected to be locally abundant in this habitat.

### 3.4. Call Properties

The manual inspection of the recording files from Mundeok revealed the presence of *D. suweonensis*, *P. nigromaculatus* and *P. chosenicus*. Analyses through R managed to extract data for *D. suweonensis* using automated detection, resulting in 89 separate call extractions from the three files the species was present in. Frequency analysis showed a mean frequency of 3.27 kHz (Figure 7). Automated and manual detection in R failed for the other two species. The call properties of the species were not different from those in the southern parts of its range [87], although the variation could not be statistically assessed due to the low sample size. 

### 3.5. Distribution Estimates and Species Richness

The results of the MaxEnt modelling highlighted a higher amphibian biodiversity and species richness in the southwest of the nation and along the northern section of the coast of the Yellow Sea (number of species ≥10; Figure 8). The lowest species richness was found in the northeast, close to the border with PR China and the Changbai and Baekdu mountains (*n* ≤ 5), while it was higher along the east coast (5 ≤ *n* ≤ 8). Generally, lower altitude and latitude were associated with higher species richness and vice versa.

Individual species distributions generally followed the same pattern (Figure 9); however, some species were more widely distributed than others, such as *B. orientalis*, *B. gargarizans*, *D. japonicus*, *G. emeljanovi*, *P. nigromaculatus* and *R. huanrenensis*. These species are also the ones with the largest described range in Northeast Asia. Other species were limited to the southwestern areas, also matching with lowlands (*D. suweonensis*, *H. leechii*, *K. borealis* and *R. coreana*). *Bufo stejnegeri*, *R. uenoi* and *O. koreanus* were remarkable due to their modelled presence in the southeast of the nation only. *Salamandrella tridactyla* was modelled as present in the northeastern region only or in high-elevation habitats. *Rana dybowskii* followed the same pattern, although reaching lower latitudes. The model for *Onychodactylus* sp. showed that the species are likely abundant in DPR Korea, although species assignment is so far unknown. The distribution models for *P. chosenicus* and *S. raddei* provided larger ranges than expected when compared to their distributions in adjacent nations [152], likely due to missing data, while the model for *R. amurensis* was very likely imperfect for the same reasons, as well as other unknown ecological variables, such as potential exclusive competition with *R. coreana*.

Based on the distribution models, we estimated the surface area based on the minimum suitable area and the 10-percentile omission area (Table 5). When comparing the top five species with the broadest estimated range for the two estimates, only *B. orientalis*, *D. japonicus* and *P. nigromaculatus* were overlapping.

### 3.6. Extinction Risk Assessment

As a general rule, biodiversity on the Korean Peninsula has declined over the past 200 years [153] and conservation actions are urgently required [154]. As of 2013, 6.3% of terrestrial land area is protected in DPR Korea, a high number when compared to some neighbouring countries, but a figure lower than the worldwide average of 12% [155]. General threats to ecosystems in DPR Korea have been shown to be mostly linked to land use [153,156], with 21% of land used for agricultural purposes and 46% preserved as forests [155]. Some areas were, however, still largely deforested at the beginning of the millennium [157], despite strong conservation policies being recently implemented for reforestation [156], maintenance of good water quality [157] and conservation and mitigation of environmental impacts [158]. On the other hand, threats because of invasive species may be lower due to trading policies [153]. It is also important to note that numerous protected areas have been designated, some under international conventions such as Ramsar and the UNESCO Man and the Biosphere Programme [153].

Threat levels to amphibians are higher than for other animals in DPR Korea [153], in line with the global pattern [27,53,54,159]. While no species is known to be endemic to DPR Korea, among the species present in the nation, *D. suweonensis*, *P. chosenicus*, *O. koreanus*, *K. koreana* and *R. uenoi* are so far described as endemic to the Korean Peninsula. Among the species listed in this study, only *D. suweonensis* and *P. chosenicus* are listed in one of the threatened categories by the IUCN, as endangered [160] and vulnerable [152], respectively. Here, we list all threats impacting the species independently of their severity. The global conservation status categories are the ones currently available and are listed for reference only. The results of our analyses for conservation status within DPR Korea are discussed below and shown in Table 5.

#### 3.6.1. *Bombina orientalis*

This species is expected to be among the most abundant amphibian species. The species is widespread both throughout the nation and mainland Northeast Asia, with apparently large and abundant populations in a large variety of habitats. In addition, no parasites were found on the seven individuals tested [98]. The principal threat is habitat modification. Recommendation for IUCN National Red List assessment (national level): least concern. Current IUCN Global Red List assessment: least concern [161].

#### 3.6.2. *Bufo gargarizans*

Despite once thought to have been introduced [29], the presence of the species in Northeast Asia demonstrates the native status of the species. The species is widespread both throughout the nation and mainland East Asia, with apparently large and abundant populations. Parasites have been found for this species but are not assessed as threatening, including cestode or nematode parasites on six individuals [98]. The principal threat is habitat modification. Recommendation for IUCN National Red List assessment (national level): least concern. Current IUCN Global Red List assessment: least concern [162].

#### 3.6.3. *Bufo stejnegeri*

The species is vulnerable under criterion B1 on the national red list, as its EOO within the nation meets the IUCN threshold, occurring in fewer than 10 threat-defined locations, with estimated ongoing decline in its: (i) extent of occurrence; (ii) area of occupancy; (iii) area, extent or quality of habitat, potentially resulting in local extirpations. The species is restricted to high elevations and is generally found in small populations. The principal threats are habitat modification, environmental pollution and climate change. Recommendation for IUCN National Red List assessment (national level): vulnerable B1ab(i,ii,iii). Current IUCN Global Red List assessment: least concern [163].

#### 3.6.4. *Dryophytes japonicus*

The species is expected to be one of the most abundant amphibian species nationwide. The species is widespread, with apparently large and abundant populations in a large variety of habitats. In addition, no parasites were found on the only individual tested [98]. The principal threats are habitat modification and environmental pollution, but climate change is known not to be a threat [164]. Recommendation for IUCN National Red List assessment (national level): least concern. Current IUCN Global Red List assessment: least concern [35].

#### 3.6.5. *Dryophytes suweonensis*

The species is Endangered under criterion B1, as it is present within four independent locations and its EOO meets the IUCN threshold due to estimated ongoing decline in its: (i) extent of occurrence; (ii) area of occupancy; (iii) area, extent or quality of habitat, potentially resulting in local extirpations. In addition, as the species is restricted to lowlands <120 m a.s.l, its range is severely fragmented. The principal threats are habitat modification [80], environmental pollution [30,165] and hybridisation [166]. There is also an increased extinction risk due to the presence of the invasive *Lithobates catesbeianus* [10], which is abundant in habitats similar to those used by the species [167] and which plays a substantial role in pathogen transmission [168]. Recommendation for IUCN National Red List assessment (national level): endangered B1ab(i,ii,ii). Current IUCN Global Red List assessment: endangered [169].

#### 3.6.6. *Glandirana emeljanovi*

The species is widespread, with apparently large and abundant populations in a large variety of habitats. The principal threat is habitat modification. Recommendation for IUCN National Red List assessment (national level): least concern. Current IUCN Global Red List assessment: least concern [170].

#### 3.6.7. *Hynobius leechii*

The species is widespread, with apparently large and abundant populations restricted to mid-elevation habitats. The principal threat is habitat modification. Recommendation for IUCN National Red List assessment (national level): least concern. Current IUCN Global Red List assessment: least concern [171].

#### 3.6.8. *Kaloula borealis*

The species is relatively widespread across the nation and in mainland Northeast Asia, although to a lesser extent than other species in the nation. The species is also susceptible to habitat degradation, which could potentially cause range contractions in the near future bringing the national EOO within the vulnerable threshold under B1 and could meet the other threshold subcriteria. The principal threats are habitat modification, chemical pollution and exploitation. Recommendation for IUCN National Red List assessment (national level): near threatened. Current IUCN Global Red List assessment: least concern [172].

#### 3.6.9. *Onychodactylus* sp.

It is not possible to assess the conservation status of species in this genus as it represents a group of several species within unknown boundaries; however, they seem relatively widespread, although restricted to high elevation habitats, which are under threat of habitat modification. Consequently, the principal threats are habitat modification, environmental pollution and climate change. Three of the species are absent from Table 5 as their presence is not yet confirmed in DPR Korea.

#### 3.6.10. *Onychodactylus fischeri*

Unconfirmed presence and predicted restricted distribution in the nation if present. Threats are likely to include habitat loss and climate change. Recommendation for IUCN National Red List assessment (national level): data-deficient. Current IUCN Global Red List assessment: least concern [173].

#### 3.6.11. *Onychodactylus zhaoermii*

Threats are likely to include habitat loss and climate change. Recommendation for IUCN National Red List assessment (national level): data-deficient. Current IUCN Global Red List assessment: not evaluated.

#### 3.6.12. *Onychodactylus zhangyapingi*

Threats are likely to include habitat loss and climate change. Recommendation for IUCN National Red List assessment (national level): data-deficient. Current IUCN Global Red List assessment: not evaluated.

#### 3.6.13. *Onychodactylus koreanus*

The species is vulnerable under criterion B1 on the national red list as its national EOO meets the IUCN threshold, it occurs in fewer than 10 threat-defined locations and due to estimated ongoing decline in its: (i) extent of occurrence; (ii) area of occupancy; (iii) area, extent or quality of habitat, potentially resulting in local extirpations. The principal threats are habitat modification, environmental pollution and climate change; however, population dynamics are not known for the species and the species could be assessed as EN under the A criterion if further research were to demonstrate that the population is declining, in view of the extremely limited expected range of the species (Table 5). Recommendation for IUCN National Red List assessment (national level): vulnerable B1ab(i,ii,ii). Current IUCN Global Red List assessment: not evaluated.

#### 3.6.14. *Pelophylax chosenicus*

The species is Vulnerable under criterion B1 on the national red list as its EOO within the nation meets the IUCN threshold, it occurs in fewer than 10 threat-defined locations and due to estimated ongoing decline in its: (i) extent of occurrence; (ii) area of occupancy; (iii) area, extent or quality of habitat, potentially resulting in local extirpations. The species is present in western areas, and while it seems to be thriving at some localities, it is restricted to lowlands and to a single type of habitat: wetlands. This makes the species vulnerable to habitat modification and environmental pollution. In addition, several subpopulations have reportedly declined and become extirpated as a result of droughts, likely linked to climate change. Recommendation for IUCN National Red List assessment (national level): vulnerable B1ab(i,ii,ii). Current IUCN Global Red List assessment: vulnerable [152].

#### 3.6.15. *Pelophylax nigromaculatus*

This species is expected to be among the most abundant amphibian species in the nation. Parasites were found on one tested individual but were not assessed as threatening (nematode) [98]. The species is widespread in DPR Korea and in Northeast Asia, with apparently large and abundant populations in a large variety of habitats. The principal threats are habitat modification and environmental pollution. Recommendation for IUCN National Red List assessment (national level): near threatened. Current IUCN Global Red List assessment: near threatened [174].

#### 3.6.16. *Strauchbufo raddei*

The species is vulnerable under criterion B1 on the national red list, as its EOO within the nation meets the IUCN threshold, it occurs in fewer than 10 threat-defined locations and due to estimated ongoing decline in its: (i) extent of occurrence; (ii) area of occupancy; (iii) area, extent or quality of habitat, potentially resulting in local extirpations; however, it is widespread throughout Northeast Asia [175] and there is no evidence of severe population decline or range contractions, meeting the IUCN thresholds at the global level. Recommendation for IUCN National Red List assessment (national level): vulnerable B1ab(i,ii,ii). Current IUCN Global Red List assessment: least concern [176].

#### 3.6.17. *Rana amurensis*

The species is limited to northern regions but is widespread in continental northern Asia, with apparently large and abundant populations in a large variety of habitats. The principal threats are habitat modification, chemical pollution, trade and exploitation. Recommendation for IUCN National Red List assessment (national level): least concern. Current IUCN Global Red List assessment: least concern [177].

#### 3.6.18. *Rana coreana*

The species is restricted to southern regions and susceptible to habitat degradation, which is projected to continue and potentially bring the EOO of the species in the nation within the vulnerable category in the near future. In addition, several subpopulations have reportedly declined and become extirpated as a result of droughts, likely linked to climate change. The principal threats are habitat modification, chemical pollution, trade and exploitation. Recommendation for IUCN National Red List assessment (national level): near threatened. Current IUCN Global Red List assessment: least concern [178].

#### 3.6.19. *Rana dybowskii*

This species is expected to be among the most abundant amphibian species in the nation. The species is limited to northern regions but is widespread in continental northern Asia, with apparently large and abundant populations in a large variety of habitats. The principal threats are habitat modification, chemical pollution, trade and exploitation. Recommendation for IUCN National Red List assessment (national level): least concern. Current IUCN Global Red List assessment: least concern [179].

#### 3.6.20. *Rana huanrenensis*

This species is expected to be among the most abundant amphibian species in the nation. The species is widespread in the nation and in continental Northeast Asia, with apparently large and abundant populations in a large variety of habitats. The principal threats are habitat modification, chemical pollution, trade and exploitation. Recommendation for IUCN National Red List assessment (national level): least concern. Current IUCN Global Red List assessment: least concern [180].

#### 3.6.21. *Rana uenoi*

The species is vulnerable under criterion B1 on the national red list, as its EOO within the nation meets the IUCN threshold, it occurs in fewer than 10 threat-defined locations and due to estimated ongoing decline in its: (i) extent of occurrence; (ii) area of occupancy; (iii) area, extent or quality of habitat, potentially resulting in local extirpations. The principal threats are habitat modification, chemical pollution, trade and exploitation. Recommendation for IUCN National Red List assessment (national level): vulnerable B1ab(i,ii,ii). Current IUCN Global Red List assessment: not evaluated.

#### 3.6.22. *Salamandrella tridactyla*

The species is restricted to high elevation habitats and is susceptible to habitat degradation, which is projected to continue and potentially bring the EOO of the species in the nation within the vulnerable category in the near future. The principal threats are habitat modification and climate change. Recommendation for IUCN National Red List assessment (national level): near threatened. Current IUCN Global Red List assessment: not evaluated.

## 4. Discussion

Our work represents major progress in the assimilation of knowledge about amphibians in DPR Korea. Through museum collections, molecular tools and field surveys, we were able to determine the presence of species at localities where species identification had previously been unclear. The combination of field surveys, literature review and molecular analyses also provided the first dataset, which was robust enough to model the distribution of the majority of amphibian species in DPR Korea. In addition, we determined the continuity in call properties between local populations and conspecific populations in other regions. Finally, the combination of all these variables, other risks and the analysis of the literature allowed us to determine the threat levels for most species.

The species richness estimate in this paper is the first that combines all described species, giving higher values than those that had previously been published. This is due to recent descriptions (e.g., [24]), additional data from the field [10] and modelling of species that are expected to be present but have not been found yet, such as *Rana huanrenensis*. In addition, some species are likely to be found in the future, such as *Karsenia koreana* [26], although this will require additional surveys and field exploration. Anecdotally, the presence of *Karsenia koreana* was mentioned during the workshop in Pyongyang in June 2019, as a researcher had been shown a picture of a very similar looking salamander by a citizen from Geumgang mountain in the southeast of the DPR Korea, within the area where the species could potentially be present. It seems important to add that *R. chensinensis* was not found to be potentially present in DPR Korea and is unlikely to be present [181]. This point of contention likely arises from shifts in taxonomy in the region. In 1972, two species were described in the Russian Far East, *R. temporaria* (currently *R. dybowskii*) and *R. cruenta* (currently *R. amurensis*; [182]), with *R. temporaria* subsequently listed as *R. chensinensis* [183,184]. While *R. chensinensis* is more closely related to *R. huanrenensis* than *R. dybowskii* [17], records were previously attributed to the wrong species and likely still are [117].

Despite some of the target species not being found during field surveys, it is not possible to confirm their absence, as demonstrated by the presence of *Bufo gargarizans* and *Rana dybowskii* in Rason in May 2018, while they were not detected in March of the same year. When species are not present in the habitat where they are expected to occur, the adequacy of the environment for their presence can be tested. This includes landscape adequacy [185], absence of urban areas [186], absence of predators [187] or adequate water quality [188]. among others. Despite the difficulty of conducting additional surveys, obtaining water quality samples from wetlands along the west coast where *Dryophytes suweonensis* and *Pelophylax chosenicus* are expected to occur would allow testing for nitrate and phosphate concentrations, as these chemicals are known to have negative impacts on the presence of these species elsewhere in their range [30].

A single *Rana* sp. egg clutch was detected in Rason in March 2018, an unexpectedly low number, as the genus was breeding during the same week in R Korea and in Russia, and the environmental variables measured were within the range of preference for the genus. Furthermore, a local farmer confirmed that “small brown frogs” are generally common in the area at the end of winter; thus, not detecting early breeding species during our surveys in Rason area in opposition to the presence of late breeding species [52] may have been related to differences in ecological requirements and behaviour. The absence of large forested landscapes in the area may be a negative point for the presence of amphibians requiring shelters to hibernate, such as *R. dybowskii*, as opposed to species that hibernate under water, such as *Pelophylax* sp. or *R. coreana* [189]. This, however, does not match with the large presence of *D. japonicus* in the area, a species that migrates to forested hills for brumation and hibernations [190].

Despite the age of samples used for molecular analyses, we were able to assess the presence of *R. uenoi* as far north as Pyongyang (Figure 1). This places the contact zone between *R. uenoi* and *R. dybowskii* somewhere between Pyongyang and the border with PR China. This pattern is relatively common for the species studied here; for instance, *R. coreana* is present in R Korea, but *R. amurensis* and *R. kukunoris* are present further north in PR China and Russia. The pattern is the same for *R. huanrenensis, Onychodactylus koreanus* and *O. zhaoermii*. This contact zone between sister clades likely results from the shift in ecological properties in the region ([55]; Figure 1): the density and presence of deciduous forests decreases along with wetness and vegetation on a northward gradient, matching with a decrease in species richness (Figure 8) and increases in Manchurian and Changbai mixed forests. We expect the species to be locally adapted to the habitat type, such as commonly seen in amphibians [191,192,193,194]. This pattern can also be generalised for other fauna, as it is also observed in mammals [195] and birds [196].

This contact zone may also have been shaped by climatic variations during ice ages, as geological and paleoclimatic variations had significant impacts on speciation [197,198,199,200]. When these variations result in the loss of connectivity between populations, such as during isolation in refugia, populations can split and diverge into segregated clades [201,202]. In the Northern Hemisphere, glacial maxima pushed most species further south, including populations in Northeast Asia [203,204], and consequently increased geographic distances between metapopulations [205,206]. These older boundaries may form the current contact zone between species, such as seen in *Hyla orientalis* in the Balkans [207], or result in population expansion from refugia, as hypothesised for raccoon dogs in north-east Asia [204]. Here, we hypothesise that this pattern of contact zones arose from climatic variations for the *Rana* complex, which diverged as a result of the orogenesis of the Changbai Mountain Range during the Miocene [42]. In addition, this orogenesis event may also have had an impact on some other amphibian species, as seen through temporally matching divergences in other amphibian genera [37,208,209].

Interestingly, the lower number of species in the northern regions of DPR Korea contrasts with species distributed around the Yellow Sea, where weaker patterns may result from the absence of elevational barriers [208,209,210,211,212,213,214,215]. Species bordering the Yellow Sea are generally distributed in lowlands and show lower interspecific variation than those present to the north and south of DPR Korea. This is demonstrated by *D. suweonensis* and *D. immaculatus* [79], which are distributed on either sides of the Yellow Sea and recently diverged [37]. Other species, such as *Kaloula borealis* [216] and *Pelophylax* sp. [208,217,218], exhibit only low genetic divergence along the same geographic gradient, a potential result of the Yellow Sea being drained of salt water repeatedly during geological ages, enabling gene exchange between the Asian mainland, the Korean Peninsula and the Japanese archipelago [219,220,221,222].

*Bufo gargarizans* is especially interesting, in that it is distributed both south and north of the Changbai Mountain Range, east and west of the Yellow Sea [162] and as far south as Vietnam [223]. It is a monophyletic species throughout its range [209,224,225,226,227,228]; however, two clades are present within *B. gargarizans* [209], and further genetic analyses may provide a different picture for this species. One of the reasons why *B. gargarizans* is not segregated into several clades like *Rana* sp. is that the recent colonisation of more northern latitudes may have followed on from a drop in sea level, connecting refugia on the Korean Peninsula and the Chinese mainland south of the permafrost line [226,229,230]. These movements would have been facilitated by the drainage basins of the Han, Amur, Liao and Yellow rivers into a single water body flowing into the ocean south of the current Yellow Sea [231,232] and by the development of the monsoon system [233,234].

In terms of species richness, the pattern found through models was in line with expected results, with fewer species at higher latitude and altitude [235,236,237,238,239,240], which is a consistent pattern in amphibians [236,241,242,243,244]. The patterns found here could be slightly different if additional data were available for Caudata, as their elevation distribution patterns are different from that of Anura on the Korean Peninsula, which are found in higher numbers at intermediate elevations, and also the species for which we have the least data available in the analyses presented here. We recommend numerous additional surveys with a specific focus on species boundaries for *Onychodactylus* sp. and the presence of *K. koreana*. Models will, however, have to take into account omission and commission errors generally associated with field surveys [134,245].

Based on available foreign reports on the ecology of DPR Korea, low population size and diversity would have been expected [153,156]; however, when considering the populations sizes of all amphibian species encountered during surveys in Mundeok, Kumya and Rason, the populations appeared to be comparatively healthy, in agreement with older [51] and relevant literature [5,9]. For instance, audio recordings of *D. suweonensis* in Mundeok (Figure 1) and *D. japonicus* in Kumya would indicate population sizes comparable to the largest ones in R Korea [36,80,81,165], while population sizes of *P. nigromaculatus*, *D. japonicus* and *B. orientalis* in Rason would likewise be comparable. This may be due to several factors, primarily the presence of large and uninterrupted agricultural wetlands, increasing the surface of available habitat matching with the ecological requirements of lowland amphibian species [246,247,248,249,250] and improving population connectivity [251,252,253]. Other reasons for large populations may be the adequacy of agricultural practices with the life cycle and behaviour of the species, where flooding matches with the breeding activities of the species and harvesting occurs after the dispersion of juveniles [254,255,256,257,258,259,260,261,262]. The comparison in population sizes is also supported by the similar morphology, as it relates to similar growth pressures, following the methodology of [263]. Furthermore, several natural wetlands connected by agricultural wetlands are now protected under the Ramsar convention, which should benefit species in the area.

In addition, the conservation status of species present at higher altitudes, such as *B. stejnegeri* and *Onychodactylus* sp., could have been improved by the designation of protected areas in DPR Korea, including biosphere reserves recognised by the UNESCO Man and the Biosphere Programme [153]. These are mostly located around Mount Baekdu and the Changbai Mountain, connected with the Biosphere Reserve in China, protecting the highest mountain on mainland Northeast Asia and also extending south along the Baekdu Mountain Range all the way to R Korea, where much of the mountain range is also included in protected area systems [153].

In term of specific threats to species and from the data gathered here, it would seem that *P. nigromaculatus*, *D. japonicus*, *K. borealis*, *B. orientalis*, *G. emeljanovi, B. gargarizans* and most *Rana* sp. are not overly threatened in DPR Korea, although small EOOs and the presence of ongoing threats do result in a threatened status for some species. The situation for *D. suweonensis*, *S. tridactyla*, *B. stejnegeri* and *S. raddei* would seem the most critical because of the small EOOs, likely even smaller AOOs and a combination of other factors, including but are not limited to habitat degradation, especially deforestation, climate change and pollution. In the case of *Onychodactylus* sp., it is currently not possible to clarify the conservation status, as there are too little data available regarding the identity of records included in the current concept.

The ecological data available from the DPR Korea are slowly increasing through bird and wetland resurveys, many of which have been conducted in recent years as part of national accession to the Ramsar Convention and formal participation in the East Asian–Australasian Flyway Partnership. Although understanding of the distribution and abundance of better-studied organisms such as birds is improving in DPR Korea, the scope and depth of such research still remains very limited compared with many other regions, and data on other species groups such as amphibians and fish are even more limited. Further field surveys combined with molecular and modelling tools have the potential to refine current the understanding of species boundaries and conservation status and to detect additional species. Such research would also be very helpful in supporting national and regional efforts to conserve biodiversity on the Korean Peninsula and throughout Northeast Asia.

## 5. Conclusions

Our results showed that species richness is higher at lower altitudes and latitudes, with up to 11 species being present. In opposition, the species richness in northern regions was half that of the lower areas. Among the amphibians species found in DPR Korea, we recommend ten species as threatened at the national level following the IUCN Red List categories and criteria. This high number is in line with threat levels to amphibians in bordering nations and globally. Species relying on agricultural wetlands such as rice paddies seem more abundant and are not under imminent threat due to the enduring presence of extensive agricultural landscapes providing adequate habitats for the species. The maintenance of such landscapes is a clear benefit to the conservation of amphibian species.

## Figures and Tables

**Figure 1 animals-11-02057-f001:**
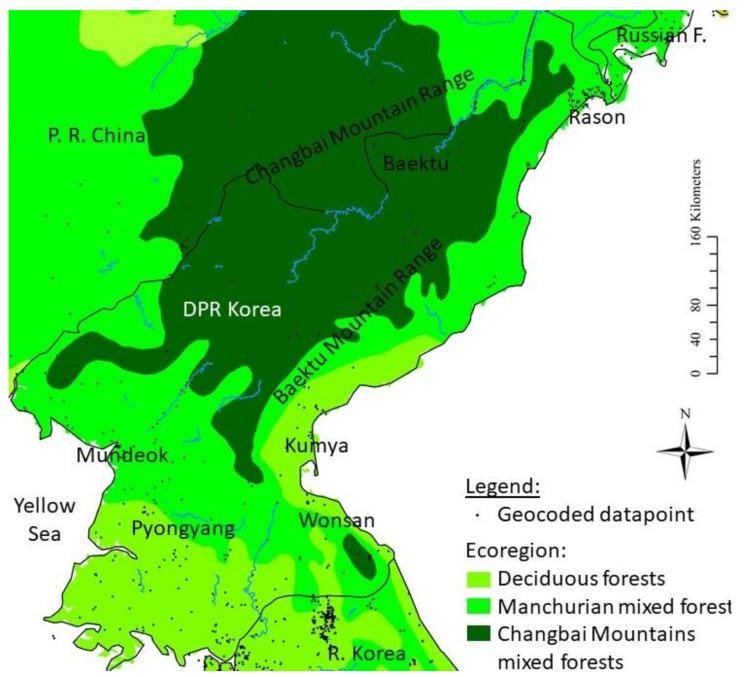
Map of survey locations and geocoded points for amphibian species presence in the Democratic People’s Republic of Korea. See the Appendix A for exact GPS coordinates for species in DPR Korea and Appendix A for all georeferenced datapoints for the species present in DPR Korea. The layer with ecoregions is attributed to [55]. Map created in ArcGIS 10.5 (ESRI; Redlands, CA, USA).

**Figure 2 animals-11-02057-f002:**
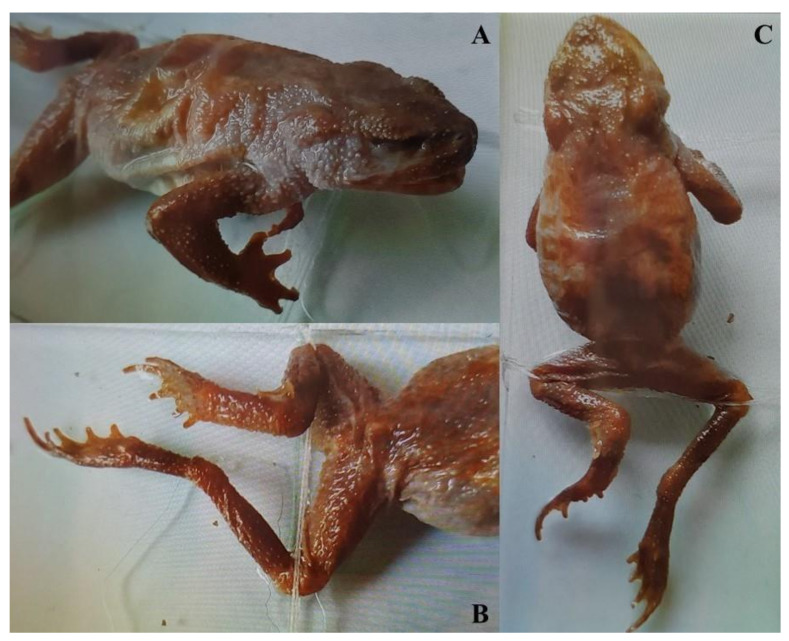
Pictures of the holotype of *Bufo sambangensis* presented by [9]. Note the morphological similarities with *B. stejnegeri*: (**A**) detailed pictures of the head; (**B**) detailed picture of the hind limbs; (**C**) dorsal picture. Pictures taken by Mr. Tu Yong Nam from Institute of Zoology of the State Academy of Science in the Democratic People’s Republic of Korea.

**Figure 3 animals-11-02057-f003:**
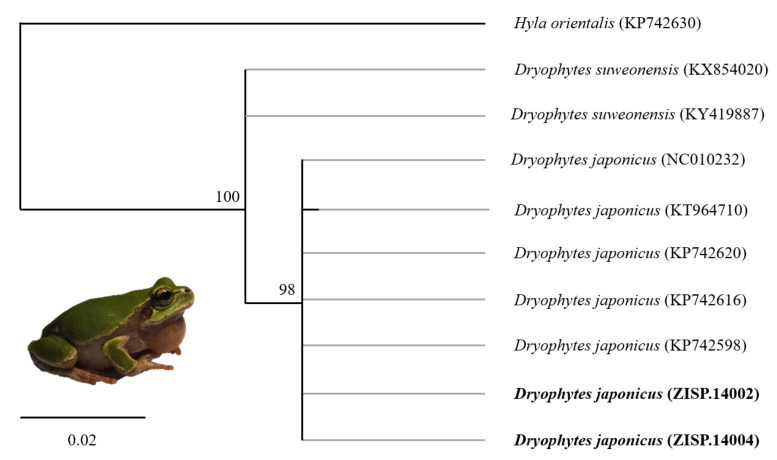
Simplified maximum likelihood phylogenetic tree built with the PHYML plugin in Geneious for *Dryophytes* sp. samples. The results clarify the presence of *D. japonicus* in the vicinity of Pyongyang, DPR Korea. *Hyla orientalis* is used as outgroup in this analysis (GenBank accession numbers shown in Table 2). Branch distances represent nucleotide substitution rates and the scale bar represents the number of changes per nucleotide position. Samples originating from this study are shown in bold.

**Figure 4 animals-11-02057-f004:**
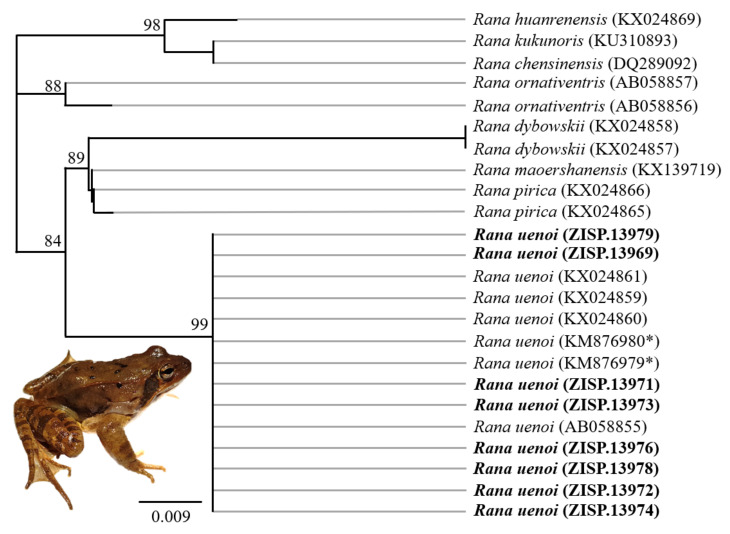
Simplified maximum likelihood phylogenetic tree built with the PHYML plugin in Geneious for *Rana* sp. samples. The results clarify the presence of *R. uenoi* in the vicinity of Pyongyang, DPR Korea (GenBank accession numbers shown in Table 2). Branch distances represent nucleotide substitution rates and the scale bar represents the number of changes per nucleotide position. The species identification for individual *Rana dybowskii* from R Korea was arbitrarily reassigned to *R. uenoi* based on the study by Matsui (2014). These individuals are indicated by *. Samples originating from this study are shown in bold.

**Figure 5 animals-11-02057-f005:**
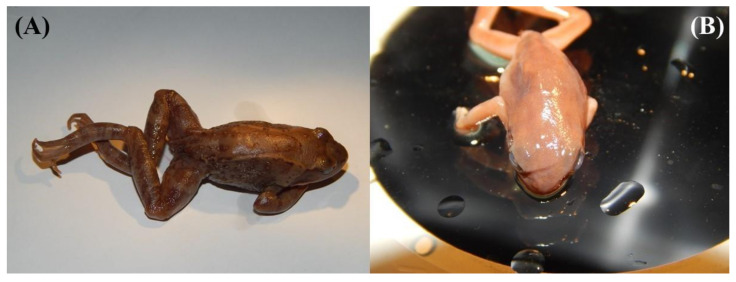
Pictures of *Rana coreana* or *Rana amurensis* (**A**) and *Dryophytes suweonensis* (**B**) used for species identification in the vicinity of Pyongyang. The samples are stored at the ZISP (Zoological Institute of the Russian Academy of Sciences, St. Petersburg, Russia).

**Figure 6 animals-11-02057-f006:**
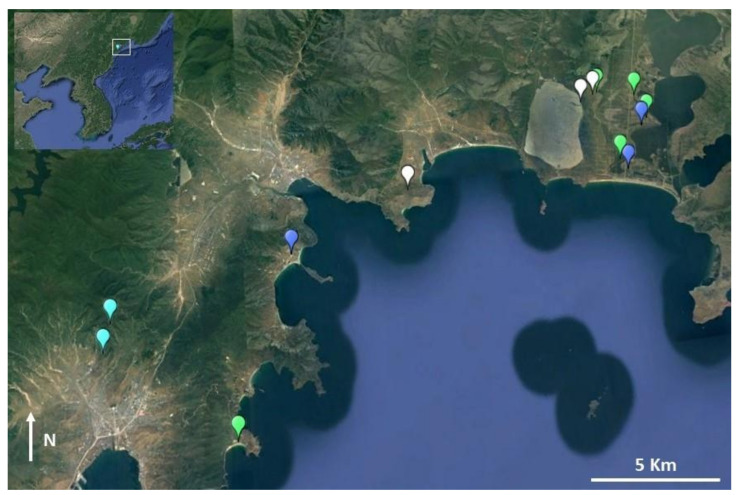
Location of the sampling sites in Rason, North Hamgyong, DPR Korea. Dark blue pins indicate agricultural wetlands, green pins indicate natural wetlands, white pins indicate grassy hills and light blue pins indicate mountainous areas. Satellite views extracted from Google Earth Pro (Google Earth Pro imagery, v7.1.2.2041; Mountain View, CA, USA).

**Figure 7 animals-11-02057-f007:**
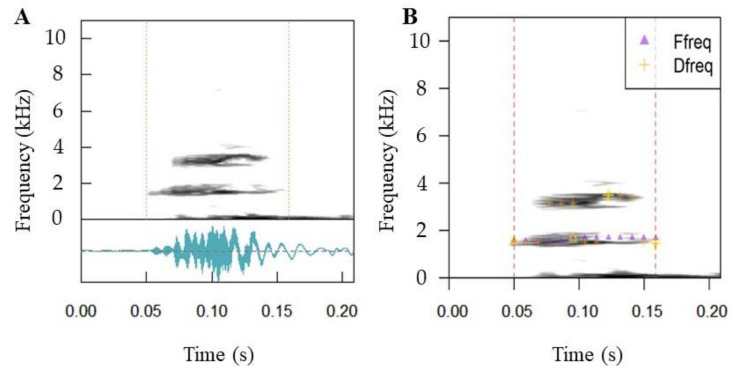
Spectral call properties for *Dryophytes suweonensis*, recorded in Mundeok, DPR Korea, in May 2018. (**A**) Spectrogram with oscillogram underneath a representative call of *D. suweonensis.* The spectrogram highlights the call between the orange vertical dashed lines. (**B**) Frequency tracking of the same *D. suweonensis* call. Calls detected are highlighted between red dashed lines. Fundamental frequency measurements are shown as purple triangles and dominant frequency measurements as yellow crosses.

**Figure 8 animals-11-02057-f008:**
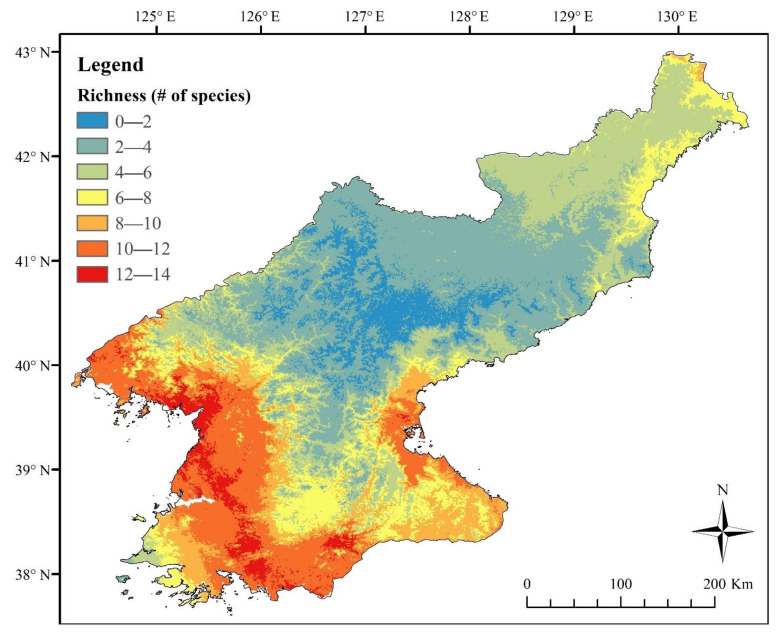
Amphibian species richness for all amphibian species recorded in DPR Korea. This map combines MaxEnt models (Figure 9) using a sum of presence probabilities. Map was created in ArcGIS 10.5 (ESRI; Redlands, CA, USA).

**Figure 9 animals-11-02057-f009:**
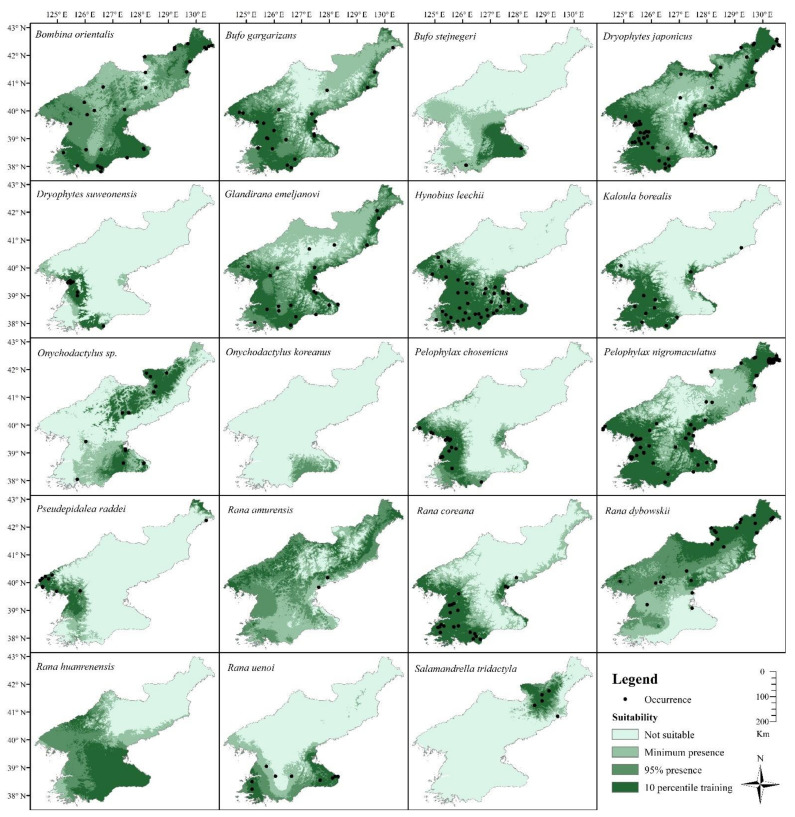
MaxEnt models for all amphibian species recorded in DPR Korea. Maps were created in ArcGIS 10.5 (ESRI; Redlands, CA, USA).

**Table 1 animals-11-02057-t001:** Amphibian collection at the Zoological Institute of the Russian Academy of Sciences, St. Petersburg, Russia, and the Zoological Museum of Moscow State University, Moscow, Russia, originating from DPR Korea and collected by the Chosun Academy of Science (DPR Korea Academy of Sciences). These datapoints are not reported later in the text in case of overlap with known combinations of species and localities.

Voucher ID	Species	Description
ZISP.2150	*Bombinator orientalis*	Korea, 1898, 2 sp., Imperial Russian Geographical Society (perhaps, A. Zvegintsov; description of the trip in Zvegintsov (1900); his route—Tumangan River, Purenga, Musan, Peyktusan volcano, Amnoka River, Kange, Togurion-Miti pass, Chin’-Chan’-Gan’ River, Tsynampo).
ZISP.2151	“*Rana* sp.”	Korea, 1898, 1sp. (lost), Imperial Russian Geographical Society (the same).
ZISP.2835	*Pelophylax nigromaculatus*	Wonsan (=Genzan), Korea, 1900, 4 sp., P.Yu. Schmidt.
ZISP.2875	*Pelophylax nigromaculatus*	Wonsan (=Genzan), Korea, 1900, 5 sp., P.Yu. Schmidt.
ZISP.2876	*Pelophylax nigromaculatus*	Wonsan (=Genzan), Korea, 5 July 1900, 8 sp., P.Yu. Schmidt.
ZISP.2912	*Bombina orientalis*	between Tonchen and Chogu-Chyen-Dogu Gulf, Korea, 1900, 4 sp., P.Yu. Schmidt
ZISP.2914	*Bufo stejnegeri*	Korea, 1900, 1 sp., P.Yu. Schmidt (description of the trip in Schmidt (1900); his route—03–24 July, Wonsan—25 July, Anbion—Suchumde Lake—27 July, Tonchen—28 July, Chogu-Chyen-Dogu Gulf—30 July, Singesa monastery, Diamond Mountains—01 August, Uonjon termal springs—02 August, Uonjon’on Pass—03 August, Chaansa monastery—04 August, Peansa monastery—05 August, Ammudo-Koge pass—05–06 August, Yuchomsa monastery—07–09 August, Kosyon Town—10–12 August, Kansen—13 August, Oridin village—14 August, Naksans monastery—15 August, Yak-Yan Mount—16 August, Kannyn Town—17 August, Kusan village—18 August, Tegulien Pass—19–20 August, Ol’chansa monastery—21 August, Kvanmul’ River—22 August, Pkhen-Khan Town—23 August, Pkhen-Khan River—24 August, Chkhe-Chkhen Mount—25 August, Iyonmusa village—26 August, Changnim village—27–28 August, Shuakori village—29 August, Chu-Yon-Che pass—30 August, Iyochen’ Town—01 August, Andon Town—02 August, Chingpo Town—04 August, Iondo Town—Khynkhe Town—Kyonju Town—12 August, Fuzan City).
ZISP.2915	*Hynobius leechii*	Korea (according to date—Ammudo-Koge pass), 05 August 1900, 2 sp., P.Yu. Schmidt.
ZISP.2925	*Bombina orientalis*	Korea, 1900, 5 sp., P.Yu. Schmidt.
ZISP.3044	*Pelophylax nigromaculatus*	rice fields in Wonsan (Genzan), Korea, 05 July 1900, 4 sp., P.Yu. Schmidt.
ZISP.13968–13981	*Rana uenoi*	Korea, 14 sp., Chosun Academy of Science.
ZISP.13982–13990	*Pelophylax nigromaculatus*	Korea, 9 sp., Chosun Academy of Science.
ZISP.13991–13995	*Pelophylax chosenicus*	Korea, 5 sp., Chosun Academy of Science.
ZISP.13996–14000	*Glandirana emeljanovi*	Korea, 5 sp., Chosun Academy of Science.
ZISP.14001–14009	*Dryophytes japonicus*	Korea, 9 sp., Chosun Academy of Science.
ZISP.14010–14011	*Hynobius leechii*	Korea, 2 sp., Chosun Academy of Science.
ZISP.14012	*Onychodactylus koreanus*	Korea, 1 sp., Chosun Academy of Science.
ZISP.14013–14016	*Bufo gargarizans*	Korea, 4 sp., Chosun Academy of Science.
ZISP.14017–14018	*Kaloula borealis*	Korea, 2 sp., Chosun Academy of Science.
ZISP.14019–14021	*Dryophytes* *suweonensis*	Pyongyang, Korea, 25–31 July 1947, 3 sp., V. Gnezdintsev.
ZISP.14022–14033	*Pelophylax nigromaculatus*	Pyongyang, Korea, 25–31 July 1947, 12 sp., V. Gnezdintsev.
ZISP.14034–14035	*Rana uenoi*	Pyongyang, Korea, 25–31 July 1947, 2 sp., V. Gnezdintsev.
ZISP.14036–14037	*Rana coreana*	Pyongyang, Korea, 25–31 July 1947, 2 sp., V. Gnezdintsev.
ZISP.14038–14044	*Kaloula borealis*	Pyongyang, Korea, 25–31 July 1947, 7 sp., V. Gnezdintsev.
ZISP.14045–14047	*Bombina orientalis*	Eiko, Korea, 29 August 1947, 3 sp., V. Gnezdintsev.
ZMMGU.871	*Pelophylax nigromaculatus*	Kheyduzuo [=Haeju], 6 October 1947, D. I. Bibikov, 1 sp.
ZMMGU.877	*Rana dybowskii*	8 October 1947, Kheyduzuo, D. I. Bibikov, 3 sp.
ZMMGU.878	*Strauchbufo raddei*	6 October 1947, Kheyduzuo, D. I. Bibikov, 6 sp.
ZMMGU.1141	*Pelophylax nigromaculatus*	Hamhung, 25 September 1970, R. Bielawski, 1 sp.

**Table 2 animals-11-02057-t002:** List of samples with GenBank accession numbers used in this analysis. The species identification for individual *Rana dybowskii* samples from R Korea was arbitrarily reassigned to *R. uenoi* based on [40]. These individuals are indicated by *. References are indicated when available or left blank.

Accession Number	Species	Locality	References
*Dryophytes*			
KY419887	*Dryophytes suweonensis*	Pyeongtaek, R Korea	[67]
KX854020	*Dryophytes suweonensis*	Pyeongtaek, R Korea	[66]
KP742630	*Hyla orientalis*	F. Russia	[65]
KP742620	*Dryophytes japonicus*	Tsushima, Japan	
KP742616	*Dryophytes japonicus*	Taksimo, F. Russia	
KP742598	*Dryophytes japonicus*	Shenyang, PR China	
NC010232	*Dryophytes japonicus*	Hiroshima, Japan	[64]
KT964710	*Dryophytes japonicus*	Heilongjiang, PR China	[68]
*Rana*			
KX024858	*Rana dybowskii*	Huanran, PR China	[42]
KX024857	*Rana dybowskii*	Fushun, PR China	[42]
KX024869	*Rana huanrenensis*	Huanran, PR China	[42]
KU310893	*Rana kukunoris*	PR China	
DQ289092	*Rana chensinensis*	Huixian, PR China	[48]
KX024861	*Rana uenoi*	Tsushima, Japan	[42]
KX024859	*Rana uenoi*	Tsushima, Japan	[42]
KX024860	*Rana uenoi*	Tsushima, Japan	[42]
KM876980	*Rana uenoi **	Boeun, R Korea	[42]
KM876979	*Rana uenoi **	Boeun, R Korea	[42]
AB058855	*Rana uenoi*	Tsushima, Japan	[42]
KX139719	*Rana maoershanensis*	Guangxi, PR China	[70]
KX024866	*Rana pirica*	Hokkaido, Japan	[42]
KX024865	*Rana pirica*	Hokkaido, Japan	[42]
AB058857	*Rana ornativentris*	Aomori, Japan	[69]
AB058856	*Rana ornativentris*	Hiroshima, Japan	[69]

**Table 3 animals-11-02057-t003:** AUC and TSS values of the 10-percentile training omission threshold generated by the MaxEnt models. The values were used to select the best fitting models.

Species	AUC ± SD	TSS ± SD
*Bombina orientalis*	0.8651 ± 0.0116	0.4626 ± 0.0805
*Bufo gargarizans*	0.8617 ± 0.0199	0.4570 ± 0.0568
*Bufo stejnegeri*	0.9387 ± 0.0078	0.6905 ± 0.1167
*Dryophytes japonicus*	0.8459 ± 0.0155	0.4958 ± 0.0525
*Dryophytes suweonensis*	0.9637 ± 0.0064	0.8148 ± 0.0767
*Glandirana emeljanovi*	0.8877 ± 0.0367	0.5080 ± 0.1041
*Hynobius leechii*	0.8459 ± 0.0134	0.4538 ± 0.0544
*Kaloula borealis*	0.9158 ± 0.0142	0.6230 ± 0.1133
*Onychodactylus* sp.	0.9185 ± 0.0147	0.5924 ± 0.1078
*Onychodactylus koreanus*	0.9568 ± 0.0095	0.6962 ± 0.1314
*Pelophylax chosenicus*	0.9451 ± 0.0054	0.7137 ± 0.0810
*Pelophylax nigromaculatus*	0.8324 ± 0.0105	0.4085 ± 0.0518
*Strauchbufo raddei*	0.9881 ± 0.0052	0.9211 ± 0.0382
*Rana amurensis*	0.9104 ± 0.0091	0.6929 ± 0.0161
*Rana coreana*	0.8697 ± 0.0119	0.4652 ± 0.1158
*Rana dybowskii*	0.9268 ± 0.0280	0.5747 ± 0.1993
*Rana huanrenensis*	0.9913 ± 0.0030	0.8411 ± 0.1978
*Rana uenoi*	0.8869 ± 0.0079	0.5874 ± 0.0315
*Salamandrella tridactyla*	0.9716 ± 0.0143	0.4488 ± 0.0909

**Table 4 animals-11-02057-t004:** Summary of population dynamics of amphibian populations in DPR Korea or closely related regions, extracted from the literature. W * indicates “workshop in Pyongyang in June 2019”. Here, [89] is cited as reported in [51].

Species	Population Dynamics	Reference
*Bufo gargarizans*	Common, reported from urban districts	[51,89]
*Strauchbufo raddei*	Uncommon or only locally abundant	[51,89]
*Dryophytes japonicus*	Present all over the nation	[51,89]
*Bombina orientalis*	Present all over the nation	[51,89]
*Glandirana emeljanovi*	One of the most common species in Central Korea	[50]
*Hynobius leechii*	Common in mountain forests	[51]
*Onychodactylus*	Common in mountain forests	[51]
*Salamandrella tridactyla*	Only in the high regions of the mountains	[51]
*Pelophylax nigromaculatus*	Abundant in rice paddies	[113]
*Rana dybowskii-R. uenoi*	Locally abundant	[113]
*Bombina orientalis*	Abundant in the lowlands	[113]
*Bufo gargarizans*	Abundant in the lowlands	[113]
*Dryophytes japonicus*	Abundant in the lowlands	[113]
*Pelophylax chosenicus*	Decline in abundance. Decline in North Pyongan province and Seoncheon because of droughts likely linked to climate change	W *
*Rana coreana*	Decline in abundance	W *
*Bufo gargarizans:*	High density on the west coast	W *
*Glandirana emeljanovi*	Not abundant but widespread	W *
*Kaloula borealis*	Continuous presence but not common, all the way to North Hamgyong province	W *
*Pelophylax nigromaculatus*	Most abundant amphibian species	W *
*Strauchbufo raddei*	Isolated populations in Shinuiju, Amrok River, Chilbeok mountain, North Pyongan province	W *
*Rana dybowskii*	Abundant at high altitude, populations >2100 m at Baekdu lake	W *
*Hynobius leechii*	Abundant <1000 m asl throughout its range	W *
*Lithobates catesbeianus*	Isolated large populations originating from escaped individuals from farms	W *
*Rana sp.*	Isolated large populations originating from escaped individuals from farms	W *

**Table 5 animals-11-02057-t005:** Description of presence models used for each species in DPR Korea. The surface area in km^2^ provided matches with the 10-percentile training and the 95% presence used in models. IUCN threat levels as of February 2019 (www.iucnredlist.org (accessed on 1 February 2019)).

Species	Minimum Suitable Area (km^2^)	10-Percentile Omission Area (km^2^)	IUCN Status	Recommendation at National Level	Local Name
*Bombina orientalis*	119,531	30,729	LC	LC	비단개구리
*Bufo gargarizans*	101,627	29,954	LC	LC	두꺼비
*Bufo stejnegeri*	45,301	11,805	LC	VU B1ab(i,ii,iii)	금강두꺼비
*Dryophytes japonicus*	114,924	51,290	LC	LC	청개구리
*Dryophytes suweonensis*	25,214	9785	EN	EN B1ab(i,ii,iii)	수원청개구리
*Glandirana emeljanovi*	107,085	30,201	LC	LC	옴개구리
*Hynobius leechii*	60,541	41,443	LC	LC	도롱룡
*Kaloula borealis*	40,571	27,098	LC	NT	맹꽁이
*Onychodactylus* sp.	57,960	20,119			
*Onychodactylus koreanus*	8637	6	DD	VU B1ab(i,ii,iii)	발톱도롱룡
*Pelophylax chosenicus*	35,908	13,810	VU	VU B1ab(i,ii,iii)	금개구리
*Pelophylax nigromaculatus*	102,328	51,779	NT	NT	참개구리
*Strauchbufo raddei*	22,026	7,775	LC	VU B1ab(i,ii,iii)	작은두꺼비
*Rana amurensis*	110,687	17,271	LC	LC	양용개구리
*Rana coreana*	54,740	26,249	LC	NT	애기개구리
*Rana dybowskii*	101,430	30,360	LC	LC	북방산개구리/기름개구리
*Rana huanrenensis*	78,533	34,021	LC	LC	계곡산개구리
*Rana uenoi*	36,066	9579	NE	VU B1ab(i,ii,iii)	
*Salamandrella tridactyla*	15,986	6294	NE	NT	합수도롱룡

## Data Availability

The data for this study are provided in the Appendix A and are available from the Mendeley Data Repository http://dx.doi.org/10.17632/z7kgyy8chp.1 (published on 28 April 2021).

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
