# Peer review of "Update on Distribution and Conservation Status of Amphibians in the Democratic People’s Republic of Korea: Conclusions Based on Field Surveys, Environmental Modelling, Molecular Analyses and Call Properties"

_animals, 2021, doi:10.3390/ani11072057_

Round 1

Reviewer 1 Report

Borzée et al. apply a diverse set of methodologies to characterize the diversity and distribution of amphibians in the Democratic People’s Republic of Korea. Using data museum specimens, molecular data, call surveys, visual/encounter surveys, and other records the team was able to identify 18 amphibian species distributed across DPR. Using ecological niche models, Borzée and colleagues provide detailed maps on the distribution of amphibians in the county. The authors also contribute national conservation status for each of the 18 amphibians.

I commend the authors for the great amount of work that went into this manuscript. I really appreciate the thoroughness, and intense labor that went into the characterization of the amphibian diversity in DPR. Characterizing the conservation status of the DPR’s is a first step in the proper management of the herpetofauna of the region. Other than a few specific changes listed below, I have no further comments to give.

Figure 1: Can you increase the size of the geocoded data points?

Line 110: Typo

Line 147: Any reason why the location is undisclosed? Was the information lost?

Line 152: Can you define soviet times? Maybe by year, as in “prior to XXXX”

Line 293: Wrong tense

Figure 2 and figure 3 can you highlight your samples?

Line 530: Temperature of what?

Line 554: Typo

579-580: What are these numbers?

Table 5: Needs definitions of IUCN levels. The table should be able to stand by itself.

Line 653: Here and throughout; assume that the readers are not familiar with the history of the region. At least mention something like “imports into DPR are restricted geographically to [country A], [B] and [C]”

Author Response

Reviewer 1

Borzée et al. apply a diverse set of methodologies to characterize the diversity and distribution of amphibians in the Democratic People’s Republic of Korea. Using data museum specimens, molecular data, call surveys, visual/encounter surveys, and other records the team was able to identify 18 amphibian species distributed across DPR. Using ecological niche models, Borzée and colleagues provide detailed maps on the distribution of amphibians in the county. The authors also contribute national conservation status for each of the 18 amphibians.

I commend the authors for the great amount of work that went into this manuscript. I really appreciate the thoroughness, and intense labor that went into the characterization of the amphibian diversity in DPR. Characterizing the conservation status of the DPR’s is a first step in the proper management of the herpetofauna of the region. Other than a few specific changes listed below, I have no further comments to give.

=> Thank you for the comments and helping improve the manuscript, each comment has been answered in details below.

 Figure 1: Can you increase the size of the geocoded data points?

=> We have slightly increased the size of the datapoints, but not too much so that overlap is limited and readers can still estimate the density of points for each area

Line 110: Typo

=> We have corrected the sentence such as: “Species present both to the south and to the north of DPR Korea are expected to be present throughout the nation”.

Line 147: Any reason why the location is undisclosed? Was the information lost?

=> These are old samples, and the information was not given, and no one from that time is still here to provide explanations. We have changed “undisclosed” to “unspecified” to address this point

Line 152: Can you define soviet times? Maybe by year, as in “prior to XXXX”

=> We have corrected the sentence as specified: “The Rana sp. collection was conducted by the North Korea Academy of Sciences (= Chosun Academy of Science) and samples (Voucher ID in Table 1) were sent to the ZISP (Zoological Institute of the Russian Academy of Sciences, St. Petersburg, Russia) during “Soviet times”, prior to the 1990’s:

Line 293: Wrong tense

=> The tense was corrected such as: “All literature citing amphibians from DPR Korea available was analysed, although some references only reported the potential presence of the species or did not provide clear georeferenced datapoints associated with specific species”.

Figure 2 and figure 3 can you highlight your samples?

=> We assume this is about Figures 3 and 4? We have modified the figures following the recommendations of the reviewer.

Line 530: Temperature of what?

=> Air and water temperatures, we have added this information in the sentence such as: “Based on surveys conducted in R Korea, P.R. China and Russia, the air and water temperatures and air humidity were adequate for spawning by early-breeding species such as: Rana sp., Hynobius sp., Salamandrella sp. and Onychodactylus sp. [150]”.

Line 554: Typo

=> Corrected such as: “where D. japonicus tadpoles were found”

579-580: What are these numbers?

=> We added this information such as: “Bombina orientalis was detected at three different sites in densities comparable to that of populations found in R Korea and Russia (n < 10).”

Table 5: Needs definitions of IUCN levels. The table should be able to stand by itself.

=> These terms are widely used, and easily searchable. Here we argue that the table stands by itself as it is now.

Line 653: Here and throughout; assume that the readers are not familiar with the history of the region. At least mention something like “imports into DPR are restricted geographically to [country A], [B] and [C]”

=> Here we argue that this information is tangential to the point raised here, and outside of the scope of this manuscript. In addition, as it is generally variables, especially following UN sanctions and the closure of the border during the pandemic, the information is unlikely to be accurate for any time period.

Reviewer 2 Report

The paper extensively reviews the data available for the distribution of Amphibia in the Democratic People's Republic of Korea and integrates them with several new records collected with a number of techniques to give an updated database for biogeographic and conservation purposes. The analysis is deep and the results are sound, so I think that the paper deserves publication after minor revision of some points indicated in the following.  

Page 3 Line 110 "are expected to present nation" is unclear, please explain.

Page 4 Line 152 "during "Soviet times" is unclear, please explain

Page 4 Line 162 please cite the source of this protocol

Author Response

Reviewer 2

The paper extensively reviews the data available for the distribution of Amphibia in the Democratic People's Republic of Korea and integrates them with several new records collected with a number of techniques to give an updated database for biogeographic and conservation purposes. The analysis is deep and the results are sound, so I think that the paper deserves publication after minor revision of some points indicated in the following. 

=> Thank you for the comments and helping improve the manuscript, each comment has been answered in details below.

Page 3 Line 110 "are expected to present nation" is unclear, please explain.

=> We have corrected the sentence such as: “Species present both to the south and to the north of DPR Korea are expected to be present throughout the nation”.

Page 4 Line 152 "during "Soviet times" is unclear, please explain

=> Following comments from both reviewers, we have corrected the sentence as specified: “The Rana sp. collection was conducted by the North Korea Academy of Sciences (= Chosun Academy of Science) and samples (Voucher ID in Table 1) were sent to the ZISP (Zoological Institute of the Russian Academy of Sciences, St. Petersburg, Russia) during “Soviet times”, prior to the 1990’s”.

Page 4 Line 162 please cite the source of this protocol

=> This is not a specific protocol but a general good-practice recommendation, as expressed in this sentence “Consequently, fixation and dehydration were unlikely to have been conducted following current recommendations”.